# Effect of COVID-19 on Routine Childhood Vaccination in Bahir Dar City, Northwestern, Ethiopia

**DOI:** 10.3390/vaccines11101569

**Published:** 2023-10-05

**Authors:** Hanna Yemane Berhane, Alemayehu Worku, Wafaie Fawzi

**Affiliations:** 1Nutrition and Behavioral Sciences Department, Addis Continental Institute of Public Health, Addis Ababa 26751/1000, Ethiopia; 2Harvard T.H. Chan School of Public Health, Boston, MA 02115, USA; mina@hsph.harvard.edu; 3Epidemiology and Biostatistics Department, Addis Continental Institute of Public Health, Addis Ababa 26751/1000, Ethiopia; alemayehu.worku@aau.edu.et; 4School of Public Health, Addis Ababa University, Addis Ababa 1176, Ethiopia

**Keywords:** childhood vaccination, COVID-19, Ethiopia

## Abstract

Despite free vaccinations for all children, Ethiopia is one of the ten countries where most children do not receive all of their basic vaccines. The COVID-19 pandemic has disrupted service delivery and utilization worldwide. In this study, we assessed the effect of the pandemic on routine childhood vaccinations in Bahir Dar, Ethiopia. The data were collected from immunization records, health system monthly reports, and interviews with vaccination professionals. The data were analyzed using interrupted time series and thematic analyses. In 6940 records covering 2018–2022, the number of vaccine doses that were delivered steadily increased except for 2021/22. Vaccine delivery consistently increased prior to the pandemic. Immediately after the first case was reported, there were some disruptions, but they were not statistically significant compared to the pre-pandemic period. In-depth interviews also confirmed this finding, showing early pandemic fear and protective measures had an impact but were not sustained. These results show that COVID-19 has had a transient but non-significant effect on childhood vaccination. Although the interruption was statistically insignificant, it could reverse decades of progress toward safeguarding children from vaccine-preventable diseases. Therefore, we must intensify our initiatives to boost childhood vaccination rates and restore pre-pandemic services to regain momentum and avoid future setbacks.

## 1. Introduction

Vaccination is one of the simplest and most cost-effective public health interventions for reducing child mortality and morbidity. The implementation of routine vaccinations for infants at different ages, from birth to 24 months, has contributed to the aversion of 37 million deaths and reduced the number of zero-dose children [1,2]. Vaccinations not only contribute to the reduction in illness, hospitalization, and premature death from vaccine-preventable diseases but also has social, economic, and productivity gains [3,4,5]; making it a crucial element for achieving the Sustainable Development Goals (SDGs).

Through the Expanded Program on Immunization (EPI), a global effort to ensure access to vaccines against diphtheria, pertussis, tetanus, poliomyelitis, measles, and tuberculosis for all children, countries have made great strides in achieving their immunization targets [6]. Every year, more than five million vaccine-preventable deaths are avoided through the effective implementation of the program [7]. In 2020, vaccine coverage has reached 83% translating to 113 million infants worldwide receiving 3 doses of the diphtheria-tetanus-pertussis (DTP3) vaccine, and 84% of children had received 1 dose of measles-containing vaccine by their second birthday [8]. Although there have been remarkable advances made in making vaccines accessible, sustaining this progress and reaching the global target of 90% vaccination coverage has not been possible, especially for developing countries [9,10].

Although the Immunization Agenda 2030 intends to eliminate inequalities in childhood immunization, the sad reality is that 60% or six out of ten unvaccinated children reside within 10 specific countries, most of which are in low-and-middle-income countries [10]. The shifting circumstances and challenges posed by natural disasters, humanitarian crises, armed conflicts, population movements, and the COVID-19 pandemic have aggravated the situation and contributed to an increase in the number of unvaccinated children [8,10], further increasing the gaps and undermining the initiatives taken to guarantee the equitable distribution of vaccines for all nations, regardless of their income level.

Globally, studies have shown that the COVID-19 pandemic has caused major disruptions to routine immunization services; as a result, there have been setbacks in eradication efforts as well as an increase in outbreaks due to reduced vaccine availability and delays in service delivery [11]. While vaccine coverage has shown some signs of improvement after the pandemic, more than 20 million children still do not receive vital lifesaving vaccines, and this means there are two million more children who have not received the vaccines in comparison to 2019 [12]. Moreover, recovery rates vary by region, with some showing rapid recovery while others, especially Sub-Saharan Africa, show little progress, contributing to frequent outbreaks of diseases which are preventable with vaccinations [11,12,13].

Ethiopia is among the ten countries where most children do not receive all of their basic vaccines [14]. The Ethiopian immunization implementation guideline recommends that all children receive one dose of Bacillus Calmette Guerin (BCG) and Oral Polio Vaccine (OPV0) at birth; DTP-HepB1-3-Hib1-3, OPV1-3, PCV1-3, Rota1-2, and IPV at 6, 10, and 14 weeks, and Measles at 9 months, free of charge [15]. Based on the 2016 Ethiopian Demographic and Health Survey, less than four out of ten children (39%) are fully vaccinated or have received all the recommended doses. Rates of children who have received no vaccinations have improved by eight percentage points, from 24% in 2005 to 16% in 2016; despite these gains, vaccination coverage in Ethiopia remains low, with huge regional variances in the percentage of children who are fully vaccinated ranging from 15% in Afar to 89% in Addis Ababa [16].

Studies in different regions of Ethiopia have revealed that vaccine uptake is associated with household-level factors such as access to media and information, antenatal and postnatal care visits, wealth, maternal age and education level, knowledge about the benefits and side effects of vaccination, and households’ distance to health centers [17,18,19]. Besides household factors, there are health-system-level barriers such as vaccine shortage, service interruption, and poor defaulter tracing and documentation [20]. In 2020, further challenges appeared in the picture as the first case of COVID-19 was reported in Ethiopia.

Although the spread of COVID-19 in Ethiopia was minimal, with 500,000 confirmed cases and 7500 fatalities [21], there were indirect impacts on health and social services. As a result of COVID-19, routine healthcare services have been disrupted; as resources and efforts have been diverted toward responding to the pandemic, and health-seeking behaviors were altered [22,23,24]. Moreover, mobility restrictions, social distancing measures, and the fear of contracting the virus within health facilities caused further disruptions. Consequently, patients deferred going to health facilities, which raised concerns that it might stall or reverse the nation’s progress towards meeting its immunization targets as well as halt progress in improving neonatal and child health [23]. In line with this, the World Health Organization’s report reveals that lockdown measures have significantly impeded the delivery of immunization services in several countries, thereby putting 80 million children at increased risk of contracting vaccine-preventable diseases [25]. The pandemic has had a multifaceted impact on healthcare provision, especially on preventive services. In many countries, routine childhood vaccines, have been interrupted, delayed, reorganized, or even suspended in some settings because dealing with the pandemic took precedence [26].

Understanding COVID-19’s impact on routine immunization services is therefore essential to planning how to move forward. In this study, we sought to understand the effect of COVID-19 on routine childhood vaccination in Bahir Dar, Ethiopia.

## 2. Materials and Methods

This study utilized a mixed methods approach, integrating both qualitative and quantitative methodologies. As part of the qualitative component, in-depth interviews were conducted with health professionals in all health centers that were conducting their routine functions during the study period. The quantitative component utilized data extracted from routine health facility reports, i.e., vaccine registry and health management information system (HMIS) reports for the period of September 2018 to July 2022, in a selected health center in Bahir Dar city, Amhara region, Ethiopia.

In Ethiopia, routine childhood vaccination is provided free of charge in all government health facilities as part of the national expanded program for immunization. Despite this, in the Amhara region, less than half of the eligible children are vaccinated, with a coverage of 46% [13]. Bahir Dar city has been selected for this study since it is the capital and is one of the largest, most populous cities of the region. Moreover, members of the research team have existing research work and have developed relationships with health centers in and around the city. Bahir Dar is located approximately 550 km from Addis Ababa, the capital of Ethiopia. Bahir Dar is among the fast-growing cities in the country, with more than 400,000 inhabitants. The city has six sub-cities [21], three public hospitals, and six public health centers.

For the record review, the research team first evaluated all the health centers in the city based on facility size, client flow, record availability, and completeness for the intended period and then purposefully selected the health center with the most complete record. After obtaining the necessary permissions, the team scanned the record books for the specific period; data were then extracted into a data extraction template developed using Epi Info software. Once the data were extracted, we performed data cleaning using STATA 14. We cross-checked outliers and inconsistencies with the scanned documents and made edits where errors were due to data entry mistakes. In cases where correction was not possible, we removed observations with missing vaccine information or specific dates. Information on all children who received their vaccine dose at the health center in the years 2018–2022 was extracted, totaling 6940. The data were used to calculate the number of doses administered annually for all vaccines, as well as to examine the number of children who received the recommended vaccines at the health center according to the year each child received the vaccination.

Furthermore, the research team collected electronic data from the health center’s central repository of monthly health management information system (HMIS) data, which compiles the aggregate number of children who have received a specific vaccine each month. Data were extracted for the four years except for the last month, which had not been completed at the time of data collection, implying that data for 47 months was accessible. The assessment of the number of children receiving the recommended vaccines was conducted following national guidelines. Children who received all of the recommended doses were considered fully vaccinated, whereas those who received less than the recommended dose or did not receive all of the vaccines were considered partially vaccinated or unvaccinated, respectively. To understand the interruption caused by COVID-19, we examined the service delivery reports of the health center for the past four years, which is the HMIS data. Using that, an interrupted time series analysis was conducted; there were 47 time points captured for the four years, and the month when the first case of COVID-19 was reported in Ethiopia, March 2020, was used to define the pre-COVID-19 time period. The analysis was conducted using STATA 14 software and using the following equation [27]:$$Y_{t}=\beta_{0}+\beta_{1}T_{t}+\beta_{2}X_{t}+\beta_{3}X_{t}T_{t}+\varepsilon_{t},$$
where Y_t_ is the number of vaccine doses delivered at time t. T_t_ is a variable indicating time (in months) at time t from the start (t = 0 month) until the end (t = 47 months) of the observation period (2018–2022). X_t_ is a dummy variable at time t coded as 0 for the periods before COVID-19 (2018–February 2020) and as 1 after COVID-19 (March 2020–2022); X_t_T_t_ is an interaction term between the time and COVID-19 dummy. ε_t_ is the error term, β_0_ estimates the baseline level of number of vaccine doses delivered at the beginning of the period, β_1_ estimates the change in the number of vaccine doses delivered until the start of the COVID-19 pandemic, β_2_ is the change in the number of vaccine doses delivered immediately after the COVID-19 pandemic, and β_3_ is the difference in the trends of the number of vaccine doses delivered before and after the pandemic (effect of COVID-19 during the pandemic).

We conducted in-depth interviews with key informants at the health center to gain a better understanding of the barriers that hinder the uptake of immunization services, with an emphasis on understanding the interruptions associated with COVID-19. In all four health centers which were providing immunization service at the time of the study, we invited EPI focal persons and staff in vaccination rooms who have worked there for at least six months and are willing to participate. Interviews were conducted with six eligible health professionals after providing detailed information about the study’s aim, approaches, and their roles in the study. They were informed about their right to participate in and/or withdraw from the study without any consequences and about confidentiality and data use and handling. Upon obtaining consent, interviews were conducted in their offices or other private locations where they felt comfortable. All the participants, that is, the EPI focal person and eligible health care provider at the health centers, were female, nurses by profession, and had work experience ranging from 3 to 12 years.

Research assistants with Master’s degrees and prior experience conducting in-depth interviews were recruited and trained for one day. Throughout the training, we placed a strong emphasis on ensuring that participants fully understood the study objectives, study guides, the critical nature of obtaining consent, and the necessary COVID-19 safety precautions. An Amharic open-ended interview guide was prepared using input from local experts and the research team members conducting the interviews. All interviews were conducted in Amharic, the local language. The interviews on average took 40 min. Detailed field notes and audio recordings were taken, which we later transcribed and translated verbatim into English for analysis. All translations were performed by a bilingual research assistant. Afterward, a member of the research team, who was also fluent in both languages, verified the transcripts by listening to the audio recordings. All translated transcripts were uploaded to a password-protected laptop which only authorized research team members could access. The transcripts were then analyzed using thematic analysis. Initially, the lead author read and re-read the transcripts, open-coded them, and organized the codes into themes in close consultation with the research team. After several iterations, the research team reached a consensus, and the preliminary analysis was then presented and discussed among peers to offer a more balanced interpretation of the findings. The findings complement those obtained from the record reviews and provide readers with explanations.

The study protocol was reviewed and has received ethical clearance from the institutional review committee of Addis Continental Institute of Public Health, reference number ACIPH/IRB/007/2022. Additionally, necessary permission letters were facilitated from the Amhara health bureau, and the selected health centers. After explaining the objectives and procedures of the study, oral informed consent was obtained before interviews were conducted. It was emphasized to participants that their participation in the study was completely voluntary and that they were free to withdraw at any time. As part of the COVID-19 precautionary measures, facemasks and hand sanitizers were provided, interviews were conducted in well-ventilated rooms, and a safe distance between the interviewer and the interviewee was maintained during data collection to limit the risk of transmission.

## 3. Results

In the years 2018–2022, a total of 6940 children received vaccination at the selected health center. The number of doses administered per year for all vaccines is variable, with the lowest number administered in 2018/19 for all vaccines except OPV0 and BCG and the highest in the year 2020/21 (Figure 1). The number of OPV0 and measles 2 vaccines administered was relatively lower compared to other vaccines, and this pattern was consistent over the four-year period.

Moreover, when examining the number of children receiving the recommended vaccines by year at the health center, the lowest number of vaccines were delivered in 2018/19, and except for birth vaccines (OPV0 and BCG), the number of doses consistently increased in subsequent years. The number of children receiving vaccines at 6, 10, and 14 weeks has increased annually, with the exception of the 2021/22 year. The results reveal a discrepancy between the number of children who receive the 6-week vaccine and those who go on to complete the full vaccination series. Specifically, the smallest difference was noted in the 2021/22 period, where 1280 children received the 6-week vaccine but only 1075 went on to complete the series. (Figure 2).

### 3.1. Interrupted Time Series Analyses

Using interrupted time series analysis (Figure 3) we examined the differences in number of vaccine doses delivered pre-COVID-19 and during the COVID-19 pandemic for each of the 47 months (time points) during the years 2018–2022. For all vaccine types except BCG (−0.07; 95% CI: −0.74, 0.59) there was a significant increase (*p* < 0.05) in the number of vaccines delivered by the health center in the period prior to the declaration of COVID-19. However, in the first month for the period immediately after the first COVID-19 case was reported in Ethiopia, which is month 19 (X19), there was some change, but the changes were not statistically significant at *p*-value 5%. With a 10% cut-off, BCG, Penta 1, PCV 1, and Rota 1 all showed marginal effects immediately after the first COVID-19 case was reported.

In the months following the declaration of COVID-19 (_X_t19), there was a slight decrease in the number of vaccine doses delivered for all vaccines. However, this change was not statistically significant, and there appeared to be a transient but non-significant decline after COVID-19. The only exception was the Measles 2 vaccine, which decreased significantly (−5.32; CI: −7.02, −3.62) at 5% significance (Table 1).

### 3.2. COVID-19 Risk Perception and Prevention Measures: Impact on Vaccination Services

During the interviews with healthcare providers, they stated that COVID-19 had affected the health-seeking patterns of their patients. This was especially true at the beginning when there were so many unknowns. They noted that even though the effects were minimal, at first, there was a lot of confusion and fear, which led to some disruptions.

“*At the beginning there might have been some effect but it did not last long… actually, even then there weren’t as such, exaggerated dropouts. Of course, for a certain time there might have been some who did; out of fear just like everybody, …but as I had told you before, mothers were wearing masks and coming in*”—I4

Aside from the fear, preventive measures taken nationally, such as restricting transportation services, have created inconvenience; however, since families were aware of the benefits of vaccines, they were not deterred. Families with young children were determined to get them vaccinated, despite obstacles such as limited access to transportation and healthcare services as reflected in the quotes below:

“*To some extent the problem of transportation has contributed to mothers not being able to come. But I believe mothers have understood the full benefits of vaccination so vaccinations have surprisingly improved from my perspective.*”—I4

“*…concerning children’s vaccinations there wasn’t anyone who hadn’t vaccinated their child because of COVID. As a matter of fact, even when all health facilities had been shut down; there were periods where vaccinations hadn’t stopped. Because of how eager the community had been to vaccinate their children.*”—I5

Additionally, at the initial stages of the pandemic, there were some changes within the health system that have led to some delays in services; for example, staff shifts were changed and staff were reallocated to other departments, which led to some disruptions, but those were resolved before they had a large impact. The health providers reported that this was a short-lived disruption and though it might have caused some disruptions they were deemed to be minimal.

“*During COVID, especially with professionals being at work at alternating times; although there was a slight lagging, interruption; we have now corrected. So, I don’t believe it has had any effects.*”—I3

As per the respondents, part of the reason for the minimal disruption in vaccine delivery can be attributed to the general misconceptions of the community; for one, many people in the community did not believe there was a pandemic in the first place. As far as people were concerned, the pandemic had been considered a ploy by the government to divert their attention and distract them from the actual situation, and they had doubts regarding its existence. There were also people who came to the health care providers for other reasons: they thought they were not susceptible or did not perceive the condition as severe, or they believed that their faith or creator would protect them.

“*They don’t believe us “It’s a political conspiracy [they would say], COVID doesn’t exist in Ethiopia.”*”—I4

“*Concerning COVID, “We have faith in our creator, who will heal us. COVID doesn’t exist in Ethiopia.” [are the comments we receive]*”—I3.

## 4. Discussion

In this study, we reviewed health facility data and found that child vaccination rates consistently increased during the period 2018–2022, except in the last year. Further, it was found that there is a discrepancy in the number of children who receive the 6-week vaccine and those who go on to be fully vaccinated. As far as the interruption caused by the COVID-19 pandemic is concerned, it was found that there was no statistically significant difference between pre-COVID-19 and during the pandemic for all vaccination types, with the exception of measles 2, which showed a significant decrease. Interviews with health providers also confirmed that although there were some disruptions at the beginning due to fear, confusion, and preventive measures being taken, they did not linger.

Vaccines delivered at the health center have increased over the years, with the exception of the last year. This increase aligns with national reports indicating a steady progress in vaccination coverage; according to the Ethiopian demographic and health survey vaccine coverage increased three-fold from 2005–2019 [28]. The reduction within the last year could be explained by the conflict in the region; which has been reported to constrain the availability of essential medications and services [29]. This was also brought up during the discussion with the health professionals, who stated that there have been disruptions in medical supplies as a result of the country’s present situation and the conflict in the areas.

When it comes to vaccine doses delivered at the health center, birth vaccines (BCG and OPV-0) are at the bottom for all years. A study from Ethiopia found that vaccine delays, particularly for the BCG vaccine, can occur for up to 2 months [30]. Additionally, the study emphasized that institutional delivery is crucial for timely vaccine delivery; highlighting that children born in health facilities are more likely to receive BCG and polio birth doses, regardless of their families’ social and economic characteristics. This could also be an explanation for our finding, which shows low birth vaccine delivery. One in five women in the study area, Bahir Dar city, gives birth at home [31]. These families may bring their children to the health center after settling down for the 6-week vaccination. Additionally, the study was taken from one health center, and in some cultures, women tend to move closer to their relatives when they are about to give birth. As a result, they might have started their vaccination elsewhere and continued here with their regular provider when they returned to their normal routine.

Our finding also showed that there is a transient but non-significant decline after COVID-19. The possibility of COVID-19 being particularly devastating to low-income countries on multiple fronts was anticipated, including but not limited to healthcare delivery, economics, and global health security [32,33]. However, the effect of COVID-19 was minimal or not statistically significant in this study. It is important to note that this does not mean that the effect did not exist. Although the number indicates a marginal effect, there are some effects that may impede the immunization program’s progress. For example, if we consider Penta I and Penta 3 and assume that service continued uninterrupted by COVID-19, we would expect to see the red line (Figure 4) on the predicted line. However, as shown in the image, the predicted line (black line) after the interruption is below the red, indicating that delivery did not continue at the same rate as before the pandemic. This means that although the effect of COVID-19 on vaccination coverage was not significant enough to be detected by the study, there are still some effects that could have impeded the progress of the immunization program.

Likewise, in the discussion with the health providers, they indicated that there were some interruptions in the early stages of the pandemic owing to public concern about contracting the disease and the preventive measures taken; however, this did not dissuade mothers from bringing their children for vaccinations. This finding is in line with findings from other parts of Ethiopia. A study from the Oromia region also reported a brief disruption, in the beginning, both on the demand side as mothers/families feared contracting the infection, and also on the supply side, as vaccine supply was temporarily interrupted at the beginning of the pandemic, but it was short-lived [34]. Another multi-region study assessing reproductive maternal and newborn health services also found no significant reduction in health seeking and service delivery; it also noted that for some services, the level of effect varied based on the region’s COVID-19 infection rates and caseload [24].

This study found that interruptions caused by the pandemic affected the delivery of Measles 2, contrary to studies in Ethiopia. However, studies in other contexts have found that coverage reductions were greater in doses administered later [35,36]. In part, this may be because they are offered separately and not as part of other services, and families have to attend health centers only for that, which can be discouraging, as opposed to earlier vaccines that were bundled with vitamin A supplements, other vaccines, and post-natal care. Furthermore, studies from different contexts have highlighted different pathways through which COVID-19 affects routine vaccinations. A study conduct in Pakistan emphasized the effect of lockdown measures on the supply and access to routine immunization services [37]. Furthermore, the lockdown limitation and subsequent preventative strategies have reduced the alternatives for engaging communities, such as outreach, affecting immunization services [38].

Although our findings did not show a statistically significant interruption due to COVID-19, the pandemic has caused some delays. This has also been documented in other contexts where the pandemic response has caused changes in healthcare systems, including the reallocation of funds, the shift of responsibilities from routine health services to COVID-19 care and prevention, and the reorganization of infrastructure and equipment [39,40]. These changes were necessary but also substantially reduced the capacity to provide essential health services across many countries, especially in low-income settings where the health sector is already underfunded and constrained. In these settings, the pandemic has limited the access to immunization and outreach services, resulting in 57 out of 66 countries postponing mass vaccination campaigns [41]. This has led to an increase in the number of vulnerable children susceptible to vaccine-preventable illnesses. Therefore, resuming essential health services and managing backlogs are critical to improving population health and achieving sustainable development goals.

To achieve this, it is essential to motivate health officials to devise innovative and adaptable strategies to regain momentum. In other settings, there have been some successful approaches that have been tried, including the nationwide ‘weekend opening hours’ strategy; the popularity of this strategy shows that a difficult situation such as the pandemic may lead to the redesigning of services to make them more accessible to the public [39]. Other strategies that have been successful in other settings include effective communication and mobilization activities, especially during a pandemic, to defuse misinformation and create a demand for services afterwards, to conduct catch-up campaigns as soon as the risk of transmission is minimized, and to use alternative venues for vaccination as soon as possible [40,42]. To regain the momentum lost in our vaccination program, it is imperative that we adopt one of these strategies and/or explore other innovative tactics. Further implementation study is required in this attempt to develop approaches that are contextually appropriate and effective.

The strength of this study is its use of multiple data sources, making it possible to triangulate findings and capture both data as well as health providers’ points of view. Moreover, the research team had years of rapport with the health center and the professionals, making it easy to communicate freely.

This study has various limitations that must be considered. One limitation of this study is that it is a facility-based study; hence, while our findings show that service delivery remains uninterrupted among individuals who have begun using the service, we do not know whether the disruptions occur in those who have not yet started using the service. Considering that studies have found poor institutional delivery due to fear of contracting disease, we may have overlooked some infants who had no interaction with the health facility, which could potentially be where the disruptions occur. Despite the fact that the selected health center is a good representative of the region, our findings may not be representative of the nation as a whole as there are wide differences across the nation. In addition, vaccination coverage in this study is based on aggregated data and not on vaccine information about individual children, which may have resulted in overestimations or underestimations.

## 5. Conclusions

The findings of this analysis suggest that, while not statistically significant, some lag has been created as a result of the pandemic. Thus, the findings may be beneficial in informing and motivating health officials to devise innovative and adaptable strategies to regain the momentum that was lost.

## Figures and Tables

**Figure 1 vaccines-11-01569-f001:**
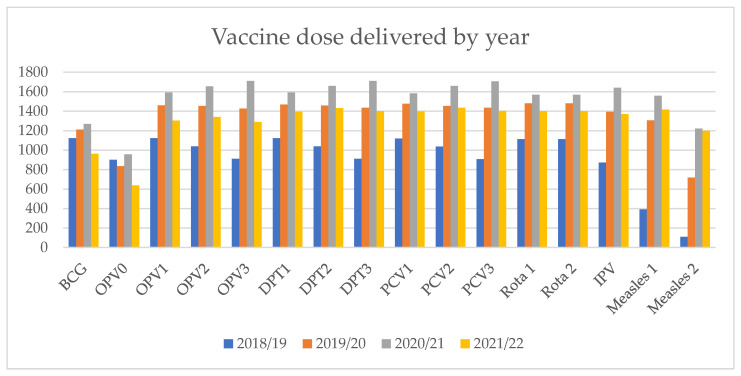
Vaccine doses delivered by year at the health center, Bahir Dar, Ethiopia.

**Figure 2 vaccines-11-01569-f002:**
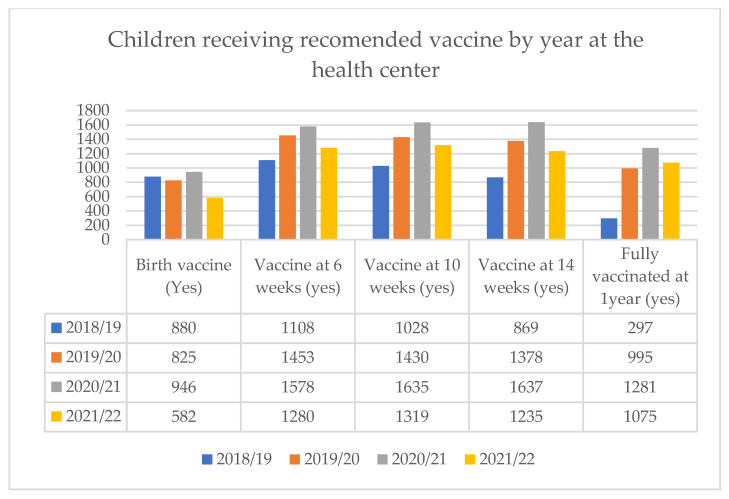
Recommended vaccine delivered by year at health center, Bahir Dar, Ethiopia.

**Figure 3 vaccines-11-01569-f003:**
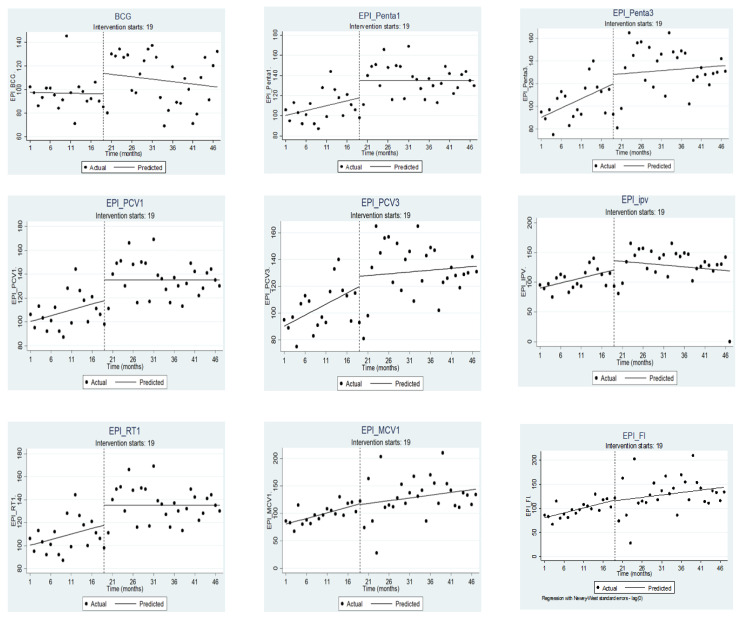
Vaccine delivered in the pre-COVID-19 and during COVID-19 periods in Bahir city, Ethiopia. NOTE: Here the vertical dashed lines indicate month 19 (March 2020), the month the first case was reported.

**Figure 4 vaccines-11-01569-f004:**
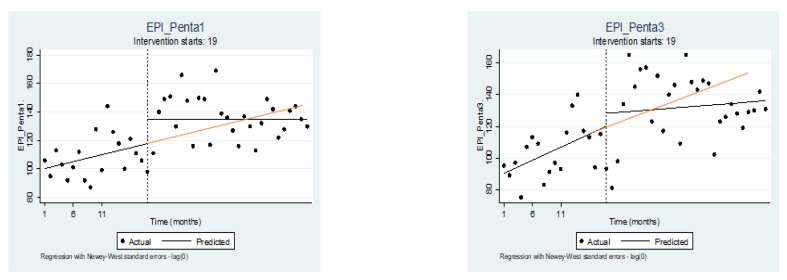
Penta 1 and Penta 3 vaccine delivered in the period before COVID-19 and during COVID-19 in Bahir city, Ethiopia. NOTE: Here the vertical dashed lines indicate month 19 (March 2020), the month the first case was reported.

**Table 1 vaccines-11-01569-t001:** Effect of COVID-19 pandemic on childhood vaccine delivery in Bahir Dar city, Ethiopia (ITSA regression with Newey–West standard errors).

	Pre-COVID-19 (_t) (Coeff: 95% CI)	Immediately after the First Case is Reported (_x19) (Coeff: 95% CI)	During the Pandemic (_x_t19) (Coeff: 95% CI)
BCG	−0.07 (−0.74, 0.59)	17.43 (−2.57, 37.43) *	−0.34 (−1.57, 0.89)
Penta 1	0.97 (0.09, 1.85) **	17.14 (−2.74, 37.03) *	−0.97 (−2.15, 0.22)
Penta 3	1.64 (0.46, 2.83) **	8.26 (−17.89, 34.41)	−1.36 (−2.97, 0.25)
PCV1	0.97 (0.09, 1.85) **	17.14 (−2.74, 37.03) *	−0.97 (−2.15, 0.22)
PCV3	1.63 (0.46, 2.83) **	7.60 (−18.47, 33.66)	−1.37(−2.97, 0.23)
IPV	1.70 (0.49, 2.91) **	15.57 (−16.72, 47.86)	−2.32 (−4.77, 0.13)
Rota 1	0.97 (0.93-1.85) **	17.14 (−2.74, 37.02) *	−0.97 (−2.15, 0.22)
Rota 2	1.20 (0.07, 2.34) **	13.88 (−11.41, 39.17)	−1.21 (−2.86, 0.43)
Measles 1	2.05 (0.86, 3.23) **	−1.47 (−36.05, 33.11)	−1.04 (−3.12, 1.03)
Measles 2	6.81 (5.93, 7.70) **	−19.95 (−50.44, 10.54)	−5.32 (−7.02, −3.62) **
Full Immunization	2.05 (0.86, 3.23) **	−1.47 (−36.05, 33.12)	−1.04 (-3.12, 1.03)

** *p*-value < 0.05 and * *p*-value < 0.1.

## Data Availability

Restrictions apply to the availability of these data. Data were obtained from a third party—immunization record and the HMIS report, which is the health center’s monthly service delivery report; therefore, we would be able to provide the data with the permission of the health center.

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
