# Peer review of "Effect of COVID-19 on Routine Childhood Vaccination in Bahir Dar City, Northwestern, Ethiopia"

_vaccines, 2023, doi:10.3390/vaccines11101569_

Round 1
Reviewer 1 Report
Researchers sought to understand the effect of COVID-19 on routine childhood vaccination in Bahir Dar, Ethiopia. It is a well-designed local study combining quantitative and qualitative data.
Minor comments:
Figure 1 is quite difficult to understand at first sight and you might think rearranging.
Please add at the limitation section that vaccination coverage is calculated using aggregated data and not on vaccine information about each child.
Author Response
Please see the attachment.
Regards,
Authors
Point-by-point Response to Reviewers’ comments
Journal: Vaccines (ISSN 2076-393X)
Manuscript ID: vaccines-2595988
Manuscript title- Effect of COVID-19 on routine childhood vaccination in Bahir Dar city, North-western, Ethiopia
Responses to Reviewers
Thank you for taking the time to review our paper. We appreciate your comments and suggestions, which helped us refine our manuscript. Below we list your comments (in bold) followed by our responses in blue font color.
Reviewer 1
Minor comments:
Figure 1 is quite difficult to understand at first sight and you might think rearranging.
- RESPONSE: In figure 1, we have changed the line graph to a bar graph; we hope it shows the contrast in the number of vaccine doses delivered in the health centers at different times.

Please add at the limitation section that vaccination coverage is calculated using aggregated data and not on vaccine information about each child
- RESPONSE: In the discussion section, we have included this limitation (lines 410-412).

Reviewer 2 Report
This paper reports a study of the effect of COVID-19 on routine childhood vaccination in Ethiopia. Overall, the analyses appear to be reasonably well done and the findings are interesting. Here are a couple of suggestions for revision of the manuscript.
First, carefully examine the text of the paper to add some minor edits to make clear the meaning/referents of your statements. For instance, in the Abstract, in the sentence "The pandemic period had significantly lower vaccine delivery than the pre-pandemic period", the meaning of your statement could be sharpened by indicating that the "lower vaccine delivery" references all child vaccinations. Similarly, in the next sentence the referent of the phrase "any significant changes" could be made more clear by adding "as compared to the pre-pandemic period".
Second, the interrupted time series analyses reported in Section 3.1 need to be supplemented with statement on the statistical models, methods, and software used to conduct these analyses. Some of this can be done in the text of the paper and some can be done in an appendix.
Third, while your statistical estimates of the before and after pandemic effects are not significant at the conventional .05 level, you also can note that 47 time points is relatively small and report the corresponding assessments of statistical significance at the .10 level.
The English is relatively good and needs only minor editing.
Author Response
Please see the attachment.
Regards,
Authors
Point-by-point Response to Reviewers’ comments
Journal: Vaccines (ISSN 2076-393X)
Manuscript ID: vaccines-2595988
Manuscript title- Effect of COVID-19 on routine childhood vaccination in Bahir Dar city, North-western, Ethiopia
Responses to Reviewers
Thank you for taking the time to review our paper. We appreciate your comments and suggestions, which helped us refine our manuscript. Below we list your comments (in bold) followed by our responses in blue font color.
Reviewer 2
Comments and Suggestions for Authors
First, carefully examine the text of the paper to add some minor edits to make clear the meaning/referents of your statements. For instance, in the Abstract, in the sentence "The pandemic period had significantly lower vaccine delivery than the pre-pandemic period", the meaning of your statement could be sharpened by indicating that the "lower vaccine delivery" references all child vaccinations. Similarly, in the next sentence the referent of the phrase "any significant changes" could be made more clear by adding "as compared to the pre-pandemic period".
- RESPONSE: In the abstract, we have made the revisions to clarify the sentence. (Lines 19-21)
Second, the interrupted time series analyses reported in Section 3.1 need to be supplemented with statement on the statistical models, methods, and software used to conduct these analyses. Some of this can be done in the text of the paper and some can be done in an appendix.
- RESPONSE: We have included a text that elaborates on the statistical procedures and software used in the analysis. (Lines 155-167)
Third, while your statistical estimates of the before and after pandemic effects are not significant at the conventional .05 level, you also can note that 47 time points is relatively small and report the corresponding assessments of statistical significance at the .10 level.
- RESPONSE: Since we only have 47-time points in our study, we checked the results at a significance level of .1 as shown in the table below. The results are similar except for the four vaccines BCG, Penta 1, PCV1, and Rota 1 which showed a marginal effect in the period immediately after the first case of COVID-19 was reported.
- To reflect significance levels of 0.05 and 0.1, we've updated Table 1. Refer to (Table 1) and (lines 236-240).
|
|
Pre-COVID 19 (_t) (Coeff: 95% CI) |
p-value |
Immediately after the first case reported (_x19) (Coeff: 95% CI) |
p-value |
During the pandemic (_x_t19) (Coeff: 95% CI) |
p-value |
|
BCG |
-0.07 (-0.74, 0.59) |
0.83 |
17.43 (-2.57, 37.43) |
0.09 |
-0.34 (-1.57, 0.89) |
0.58 |
|
Penta 1 |
0.97 (0.09, 1.85) |
0.03 |
17.14 (-2.74, 37.03) |
0.09 |
-0.97 (-2.15, 0.22) |
0.11 |
|
Penta 3 |
1.64 (0.46, 2.83) |
0.008 |
8.26 (-17.89, 34.41) |
0.53 |
-1.36 (-2.97, 0.25) |
0.10 |
|
PCV1 |
0.97 (0.09, 1.85) |
0.031 |
17.14 (-2.74, 37.03) |
0.09 |
-0.97 (-2.15, 0.22) |
0.12 |
|
PCV3 |
1.63 (0.46, 2.83) |
0.008 |
7.60 (-18.47, 33.66) |
0.56 |
-1.37( -2.97, 0.23) |
0.10 |
|
IPV |
1.70 (0.49, 2.91) |
0.007 |
15.57 (-16.72, 47.86) |
0.34 |
-2.32 (-4.77, 0.13) |
0.06 |
|
Rota 1 |
0.97 (0.93-1.85) |
0.03 |
17.14 (-2.74, 37.02)) |
0.09 |
-0.97 (-2.15, 0.22) |
0.11 |
|
Rota 2 |
1.20 (0.07, 2.34) |
0.04 |
13.88 (-11.41, 39.17) |
0.27 |
-1.21 (-2.86, 0.43) |
0.14 |
|
Measles 1 |
2.05 (0.86, 3.23) |
0.001 |
-1.47 (-36.05, 33.11) |
0.93 |
-1.04 (-3.12, 1.03) |
0.32 |
|
Measles 2 |
6.81 (5.93, 7.70) |
0.0001 |
-19.95 (-50.44, 10.54) |
0.19 |
-5.32 (-7.02, -3.62) * |
0.0001 |
|
Full Immunization |
2.05 (0.86, 3.23) |
0.001 |
-1.47 (-36.05, 33.12) |
0.93 |
-1.04 (-3.12, 1.03) |
0.32 |

Reviewer 3 Report
The manuscript presents interesting analysis on the trends of children vaccination in one health center from Bahir Dar city (Ethiopia). The manuscript is well written, presenting adequately the context of the study and the development of data collection and analysis. There are only small corrections that are required for improvement of the manuscript, especially referring to typos/errors:
(1) The sentence in page 2 (lines 53-55) seems to be part of the preceding phrase, thus, it would be better to join them;
(2) Minor typos/errors: "health-seeking behavior were altered" (page 2, lines 76-77); "routine childhood vaccines, have been interrupted" (page 2, line 86); "recieving" (page 5, Figure 2); "or the period immediately" (page 5, line 192); "in the region; which has been" (page 8, line 262); "early disruptions because of fears, preventive measures" (page 8, line 296);
(3) I suggest to mention that it is a "mixed methods study" before explaining that the study encompasses qualitative and quantitative approaches;
(4) Indicate why the initial period of data collection was September 2018, instead of August 2018, which would support an analysis of 48 months instead of 47 months;
(5) Figure 1 should be changed to graph presenting bars (frequencies) for each vaccine for each year, instead of presenting lines connecting the vaccination rates of different vaccines in a certain year (i.e., does not make sense to connect the frequencies of vaccination of different vaccines in a certain period);
(6) Figure 2 should present the proportion of children vaccinated with the vaccine recommended for the age bracket, instead of presenting the absolute frequency of children vaccinated;
(7) Substitute the term "clients" (page 6, line 208), instead use "patients";
(8) It would be interesting to include in the results: one table with descripttive statistics of children vaccinated per year (according to sex, age bracket, etc.), and one table with descriptive statistics of the health center staff interviewed (according to sex, age bracket, profession, etc.);
(9) Authors mention that "Although our findings did not show a statistically significant interruption due to COVID-19, the pandemic has caused some delays"; however, the study does not present data to support that sentence. It could be interesting to present the comparison between age recommended for each vaccine and age that the children was effectively being vaccinated.
(10) The study is very interesting, and contributes with substantial information in public health regarding vaccination of children, congratulations to the authors.
The quality of English language is very good with few corrections required to improve the manuscript.
Author Response
Please see the attachment.
Regards,
Authors
Point-by-point Response to Reviewers’ comments
Journal: Vaccines (ISSN 2076-393X)
Manuscript ID: vaccines-2595988
Manuscript title- Effect of COVID-19 on routine childhood vaccination in Bahir Dar city, North-western, Ethiopia
Responses to Reviewers
Thank you for taking the time to review our paper. We appreciate your comments and suggestions, which helped us refine our manuscript. Below we list your comments (in bold) followed by our responses in blue font color.
Reviewer 3
Minor comments
(1) The sentence in page 2 (lines 53-55) seems to be part of the preceding phrase, thus, it would be better to join them;
- RESPONSE: We believe it is best to keep the two paragraphs separate. The first provides global context while the second contains specific country-level details.
(2) Minor typos/errors: "health-seeking behavior were altered" (page 2, lines 76-77); "routine childhood vaccines, have been interrupted" (page 2, line 86); "recieving" (page 5, Figure 2); "or the period immediately" (page 5, line 192); "in the region; which has been" (page 8, line 262); "early disruptions because of fears, preventive measures" (page 8, line 296);
- RESPONSE: We have made edits to the corresponding texts
- "Health-seeking behavior was altered" (page 2, lines 76-77) and "routine childhood vaccines, have been interrupted" (page 2, line 86). - We did not make any changes as we were unsure of the suggestion.
- "Recieving” (page 5, Figure 2)- spelling corrected see (page 6, figure 2)
- "or the period immediately" (page 5, line 192)- we have edited the sentence to clearly indicate which period it is referring to. (lines 236-237)
- "in the region; which has been" (page 8, line 262)- we have edited the sentence (Lines 307-308)
- "early disruptions because of fears, preventive measures" (page 8, line 296)- we have made revisions to this statement see lines 345-348.
(3) I suggest to mention that it is a "mixed methods study" before explaining that the study encompasses qualitative and quantitative approaches;
- RESPONSE: As suggested, we updated the methods section to reflect the use of a mixed-method approach. (Lines 110-111)
(4) Indicate why the initial period of data collection was September 2018, instead of August 2018, which would support an analysis of 48 months instead of 47 months;
- RESPONSE: In Ethiopia, our calendar is different, and our new year falls in September. Therefore, we are reporting September instead of August because health service documents are reported in the Ethiopian calendar, and the conversion corresponds to September.
(5) Figure 1 should be changed to graph presenting bars (frequencies) for each vaccine for each year, instead of presenting lines connecting the vaccination rates of different vaccines in a certain year (i.e., does not make sense to connect the frequencies of vaccination of different vaccines in a certain period);
- RESPONSE: We agree, we have now changed Figure 1 to be a bar graph. (Figure 1)
(6) Figure 2 should present the proportion of children vaccinated with the vaccine recommended for the age bracket, instead of presenting the absolute frequency of children vaccinated;
- RESPONSE: In order to determine the proportion, it is necessary to have an estimate of the number of eligible children for each vaccine. Unfortunately, such an estimate is not available, so we have presented the absolute number instead to illustrate the variation by year. We acknowledge that this is one of the limitations of the study.
(7) Substitute the term "clients" (page 6, line 208), instead use "patients";
- RESPONSE: We have substituted. (Line 254)
(8) It would be interesting to include in the results: one table with descripttive statistics of children vaccinated per year (according to sex, age bracket, etc.), and one table with descriptive statistics of the health center staff interviewed (according to sex, age bracket, profession, etc.);
- RESPONSE: We didn't initially conduct a sex-segregated analysis because we didn't anticipate any gender differences. However, to confirm this assumption, we did conduct the analysis, and the results were as we expected - an even distribution between males and females. Thus, we do not think this table would add any value to the manuscript, however, we are open to adding this if the Reviewer and Editor feel it is important.
|
Male (#) |
% |
Female (#) |
% |
|
|
BCG |
2276 |
50.48 |
2233 |
49.52 |
|
OPV0 |
1680 |
51.13 |
1606 |
48.87 |
|
OPV1 |
2699 |
49.85 |
2715 |
50.15 |
|
OPV2 |
2721 |
50.18 |
2701 |
49.82 |
|
OPV3 |
2597 |
49.25 |
2676 |
50.75 |
|
DPT1 |
2756 |
49.98 |
2758 |
50.02 |
|
DPT2 |
2763 |
50.04 |
2759 |
49.96 |
|
DPT3 |
2658 |
49.29 |
2735 |
50.71 |
|
PCV1 |
2756 |
50.02 |
2754 |
49.98 |
|
PCV2 |
2762 |
50.05 |
2757 |
49.95 |
|
PCV3 |
2658 |
49.36 |
2727 |
50.64 |
|
Rota 1 |
2752 |
50.08 |
2743 |
49.92 |
|
Rota 2 |
2739 |
50.07 |
2731 |
49.93 |
|
IPV |
2571 |
49.32 |
2642 |
50.68 |
|
Measles 1 |
2296 |
49.71 |
2323 |
50.29 |
|
Measles 2 |
1614 |
50.23 |
1599 |
49.77 |
- RESPONSE: The description of the health providers included in the in-depth interviews is included in the text under the methods section. (Lines 179-181)
(9) Authors mention that "Although our findings did not show a statistically significant interruption due to COVID-19, the pandemic has caused some delays"; however, the study does not present data to support that sentence. It could be interesting to present the comparison between age recommended for each vaccine and age that the children was effectively being vaccinated.
- RESPONSE: "Although our findings did not show a statistically significant interruption due to COVID-19, the pandemic has caused some delays"- The statement refers to the data presented in Table 1 and Figure 3, which demonstrates that vaccine delivery was consistently increasing for all types of vaccines until the pandemic disrupted the trend. Despite not showing a statistically significant effect in our study, the images clearly indicate that the trajectory deviated from its previous pattern. The estimates provided in the table also support this observation.
- RESPONSE: For the second suggestion, which is: “ It could be interesting to present the comparison between the age recommended for each vaccine and the age that the children were effectively being vaccinated”- this suggestion is very interesting and the researchers plan to explore timeliness of vaccine in a separate analysis.

Reviewer 4 Report
This is an interesting paper on vaccination in Ethiopia. Before it can be published, several changes should be made.
Abstract:
1. "COVID-19 has had a transient but non-significant effect on childhood vaccination." This statement can be confusing. Earlier, it's stated that the pandemic had significantly lower vaccine delivery than the pre-pandemic period, which contradicts the "non-significant effect" statement. The distinction between transient effects and statistical significance needs clarification.
2. The ending statement about the program possibly losing momentum could benefit from a little more explanation. This might help drive the point home about the implications of such interruptions, even if they're statistically insignificant.
3. Statistical Analysis Mention: state explicitly in the abstract that you used "interrupted time series analysis".
Introduction:
4. Global Vaccination Trends: Before diving into the specifics of Ethiopia, it's essential to provide readers with a broader context. Begin the introduction by discussing
a. global trends in childhood vaccination. This can help set the stage for how Ethiopia compares in this global scenario.
b. Vaccination Trends in Africa: Following the global overview, provide insights specific to the African continent:
c. Impact of COVID-19 on Vaccination: After the initial context setting, you can delve into the specifics of how COVID-19 has impacted these efforts globally , within Africa, and in Ethiopia:
5. "The implementation of routine vaccinations for infants at different ages; from birth to 24 months, has contributed..." - Change the semicolon to a comma.
6. "Ethiopia is amongst the ten countries where most children do not receive all of their basic vaccines." - "Amongst" could be changed to "among" for simplicity.
7. Clarify the relation between the global situation and Ethiopia's context regarding vaccination. The introduction seems to jump back and forth between the global context and Ethiopia. Streamlining the narrative might make the context clearer for the reader.
8. In discussing the impacts of COVID-19 on healthcare in Ethiopia, give a brief overview or statistical data to show the magnitude of COVID's spread in Ethiopia, especially in Bahir Dar.
9. Mention the specific objectives of your study at the end of the introduction.
MATERIAL AND METHODS
10. Study Design Clarity: It's mentioned that both qualitative and quantitative approaches were used. It would be beneficial to have a brief outline at the beginning of the section, describing the main steps to help readers get an overview of the study design.
11. Selection Process for Health Center: The process for selecting the health center was described as purposeful based on several criteria. It may be beneficial to describe why those specific criteria were chosen. Were there other health centers that met the criteria but were excluded? If so, why?
12. Qualitative Data Collection: More specifics on the qualitative component are needed. How many in-depth interviews were conducted in total?What was the average duration of each interview? Were any participants reluctant to speak, and how was this handled? While it's mentioned that field notes and audio recordings were taken, was there any method to ensure consistency across all interviews?
13. Data Analysis: For the quantitative data, it might be useful to state if there were any cleaning or preprocessing steps, especially given data was sourced from physical records. For the qualitative data, how were themes derived in the thematic analysis? Was any software used? (e.g., Atlas Ti).
14. Informed Consent: The fact that oral informed consent was obtained is mentioned. However, considering the importance of this step, more details might be helpful. Were participants given a document outlining the study objectives and their rights, or was everything communicated orally? How was oral consent documented?
15. COVID-19 Precautions: Stating that "All COVID-19 precautionary measures were followed" is a bit generic. Given the relevance of the topic and the global interest in COVID-19 protocols, a more detailed description of the precautions taken could be provided.
16. Language and Translation: When the interviews were translated from Amharic to English, were any steps taken to ensure the translations maintained the original sentiment and nuance of the responses? Were any measures taken to validate the accuracy of the translations?
17. Statistical Analysis: Interrupted Time Series Analysis: More details on how the analysis was done might be beneficial. What statistical software was used? Were there any controls or confounders considered in the model? Was seasonality considered, given it's a four-year span.
RESULTS
18. The "Results" section should include detailed title or subheadings, such as "Vaccination Trends 2018-2022" or "Impact of COVID-19 on Vaccine Delivery".
19. Figure References and Details: When mentioning figures (e.g., Figures 1, 2, or 3), it might be useful to briefly describe the content of the figures for readers who may not have immediate access to the figures or may not know how to interpret them. Ensure that the figures have clear label legends and are self-explanatory. (A figure or table should be understood without reading the paper.
20. In Table 1 in vaccine Measles 2 vaccine, there is a note "*", but it isn't explained anywhere.
21. Qualitative Data Interpretation: The interview excerpts provide a good qualitative overview. However, after each excerpt, it might be helpful to provide a sentence summarizing or analyzing the key takeaway for the reader.
22. Consider organizing the qualitative data into themes or categories for clarity. This will help readers understand the broader conclusions being drawn from the personal accounts.
23. The paragraph starting with "Additionally, at the initial stages of the pandemic..." suggests that service disruptions were quickly addressed. Still, it's unclear how long these disruptions lasted or how they were addressed.
24. Similarly, statements like "many people in the community did not believe there was a pandemic" could benefit from quantification. Was this the majority of the community or a vocal minority? If this is based on the qualitative interviews, it should be specified.
25. Conclude the Results section with a brief summary statement that encapsulates the key findings, tying the quantitative and qualitative data together.
DISCUSSION
26. The discussion about the delivered Figure 4, Penta 1, and Penta 3 vaccines is somewhat confusing. Simplify or break it down further to make it more understandable for readers unfamiliar with the topic.
27. The limitations section is important and should be highlighted appropriately. Consider starting a new paragraph with a statement, "There are several limitations to this study that need to be addressed."
28. The conclusion is tucked at the end without a clear break or heading. Consider starting a new paragraph with a statement like "In conclusion," to clearly demarcate it from the rest of the discussion.
29. Discuss the practical implications of your findings. How might policymakers, healthcare professionals, or communities use this information?
30. While the study's strengths are mentioned, elaboration on why using multiple data sources and having a rapport with the health center are significant. This will bolster the credibility of your study.
31. There are some minor grammar and phrasing issues, e.g., "...noted for some services the level of effect varied by the regions COVID-19 infection and caseload [20]." might be clearer as "...noted that for some services, the level of effect varied based on the region's COVID-19 infection rates and caseload [20]."
32. Provide Potential Solutions: The section on regaining momentum after the pandemic would benefit from more detailed, actionable solutions or recommendations.
BIBLIOGRAPHY
33. Update the bibliography. The obsolesce index of the paper is 3 years ( publication date – median references), which is quite high for a topic so actual like this (the median of your references is 2020) , Do an additional bibliographic search and incorporate the latest references on trends and effect of covid in vaccination in the world, Africa, and Ethiopia.
Author Response
Please see the attachment.
Regards,
Authors
Point-by-point Response to Reviewers’ comments
Journal: Vaccines (ISSN 2076-393X)
Manuscript ID: vaccines-2595988
Manuscript title- Effect of COVID-19 on routine childhood vaccination in Bahir Dar city, North-western, Ethiopia
Responses to Reviewers
Thank you for taking the time to review our paper. We appreciate your comments and suggestions, which helped us refine our manuscript. Below we list your comments (in bold) followed by our responses in blue font color.
Reviewer 4
Abstract:
- "COVID-19 has had a transient but non-significant effect on childhood vaccination." This statement can be confusing. Earlier, it's stated that the pandemic had significantly lower vaccine delivery than the pre-pandemic period, which contradicts the "non-significant effect" statement. The distinction between transient effects and statistical significance needs clarification.
RESPONSE: In the Abstract, we have revised the statement above to explicitly explain that there were some variations in vaccination administration in the month after the first case was reported and subsequently, but the changes were not statistically significant. (Lines 19-21)
- The ending statement about the program possibly losing momentum could benefit from a little more explanation. This might help drive the point home about the implications of such interruptions, even if they're statistically insignificant.
RESPONSE: We have modified the statement to elaborate on the implications of such interruptions. (Lines 26-32)
- Statistical Analysis Mention: state explicitly in the abstract that you used "interrupted time series analysis".
RESPONSE: That is included in line 17.
Introduction:
- Global Vaccination Trends: Before diving into the specifics of Ethiopia, it's essential to provide readers with a broader context. Begin the introduction by discussing
- global trends in childhood vaccination. This can help set the stage for how Ethiopia compares in this global scenario.
- Vaccination Trends in Africa: Following the global overview, provide insights specific to the African continent:
- Impact of COVID-19 on Vaccination: After the initial context setting, you can delve into the specifics of how COVID-19 has impacted these efforts globally , within Africa, and in Ethiopia:
RESPONSE: We have included a paragraph that incorporates the above suggestions. (Lines 63-71)
- "The implementation of routine vaccinations for infants at different ages; from birth to 24 months, has contributed..." - Change the semicolon to a comma.
RESPONSE: The semicolon is changed to a comma. (Line 38)
- "Ethiopia is amongst the ten countries where most children do not receive all of their basic vaccines." - "Amongst" could be changed to "among" for simplicity.
RESPONSE: The suggested revision has been made on Line 72.
- Clarify the relation between the global situation and Ethiopia's context regarding vaccination. The introduction seems to jump back and forth between the global context and Ethiopia. Streamlining the narrative might make the context clearer for the reader.
RESPONSE: The format of the introductory section has been revised to reflect the recommendations made.
- In discussing the impacts of COVID-19 on healthcare in Ethiopia, give a brief overview or statistical data to show the magnitude of COVID's spread in Ethiopia, especially in Bahir Dar.
RESPONSE: We have highlighted some of the reported impacts of COVID-19 on healthcare in Ethiopia, as well as its magnitude of spread. (Lines 92-98)
- Mention the specific objectives of your study at the end of the introduction.
RESPONSE: The objective of this study is to understand the effect of COVID-19 on routine childhood vaccination in Bahir Dar, Ethiopia. (Lines 107-108)
MATERIAL AND METHODS
- Study Design Clarity: It's mentioned that both qualitative and quantitative approaches were used. It would be beneficial to have a brief outline at the beginning of the section, describing the main steps to help readers get an overview of the study design.
RESPONSE: We amended the methods section to emphasize that this study employed a mixed-method approach, as well as a follow-up discussion of what each technique entails. (Lines 110-116)
- Selection Process for Health Center: The process for selecting the health center was described as purposeful based on several criteria. It may be beneficial to describe why those specific criteria were chosen. Were there other health centers that met the criteria but were excluded? If so, why?
RESPONSE: As previously stated, we used purposive sampling to select the health center. We first evaluated all health centers in the city based on the following criteria: facility size, client flow, record availability, and completeness for the intended period, then purposefully selected the health center with the most complete record. Using this criterion, the chosen health center had the most full and well-documented record for our review.
- Qualitative Data Collection: More specifics on the qualitative component are needed. How many in-depth interviews were conducted in total? What was the average duration of each interview? Were any participants reluctant to speak, and how was this handled? While it's mentioned that field notes and audio recordings were taken, was there any method to ensure consistency across all interviews?
RESPONSE: We revised the methods section to include additional information about the qualitative component. The total number of interviews done was 6, with an average duration of 40 minutes. Willing participants from the vaccination unit were interviewed in a private setting; to guarantee consistency, we trained the research assistants, gave interview guidelines, and supervised the procedure throughout. (Lines 170-178)
- Data Analysis: For the quantitative data, it might be useful to state if there were any cleaning or preprocessing steps, especially given data was sourced from physical records. For the qualitative data, how were themes derived in the thematic analysis? Was any software used? (e.g., Atlas Ti).
RESPONSE:
- For the record review, we have included details on how the data cleaning was done. (Lines 132-135)
- The qualitative analysis was not software-assisted. The approach involves reading and re-reading transcripts, coding them, and organizing codes into themes with input from the research team. The preliminary analysis was then presented and discussed among peers to offer a more balanced interpretation of the findings. The methods section is updated to reflect these changes. (Lines 195-200)
- Informed Consent: The fact that oral informed consent was obtained is mentioned. However, considering the importance of this step, more details might be helpful. Were participants given a document outlining the study objectives and their rights, or was everything communicated orally? How was oral consent documented?
RESPONSE: The process of obtaining consent from participants included both approaches; we had a consent form prepared that outlined detailed information about the study's aim and approaches, and once that was explained to the participants, the research assistants addressed any follow-up questions. The subjects then gave their verbal agreement, and the research assistants confirmed it with their signatures. This was done due to cultural difficulties in which individuals are hesitant to sign paperwork.
- COVID-19 Precautions: Stating that "All COVID-19 precautionary measures were followed" is a bit generic. Given the relevance of the topic and the global interest in COVID-19 protocols, a more detailed description of the precautions taken could be provided.
RESPONSE: The study participants were provided with masks and sanitizers. Interviews were conducted in well-ventilated spaces with safe distancing between interviewer and interviewee. (Lines 207-211)
- Language and Translation: When the interviews were translated from Amharic to English, were any steps taken to ensure the translations maintained the original sentiment and nuance of the responses? Were any measures taken to validate the accuracy of the translations?
RESPONSE: A member of the study team who was also proficient in both languages checked the transcripts by listening to the audio recordings. We have included information on the steps taken. (Lines 190-192)
- Statistical Analysis: Interrupted Time Series Analysis: More details on how the analysis was done might be beneficial. What statistical software was used? Were there any controls or confounders considered in the model? Was seasonality considered, given it's a four-year span.
RESPONSE: We have included a text that elaborates on the statistical procedures and software used in the analysis. (Lines 145-157)
RESULTS
- The "Results" section should include detailed title or subheadings, such as "Vaccination Trends 2018-2022" or "Impact of COVID-19 on Vaccine Delivery".
- RESPONSE: We appreciate the suggestions, but we feel that having too many headings and sub-headings might be confusing, therefore we would like to retain the headers as they are.
- Figure References and Details: When mentioning figures (e.g., Figures 1, 2, or 3), it might be useful to briefly describe the content of the figures for readers who may not have immediate access to the figures or may not know how to interpret them. Ensure that the figures have clear label legends and are self-explanatory. (A figure or table should be understood without reading the paper.
- RESPONSE: We have changed the wording and/or figures type to make them clearer.
- In Table 1 in vaccine Measles 2 vaccine, there is a note "*", but it isn't explained anywhere.
- RESPONSE: The “**” in the table indicates that the Measles 2 vaccine delivery decreased significantly (-5.32; CI: -7.02, -3.62) at 5% significance. Which is also reported in the text Lines 246-247
- Qualitative Data Interpretation: The interview excerpts provide a good qualitative overview. However, after each excerpt, it might be helpful to provide a sentence summarizing or analyzing the key takeaway for the reader.
- RESPONSE: We attempted to summarize the main points in the text above the quotations; would it be beneficial to flip the order?
- Consider organizing the qualitative data into themes or categories for clarity. This will help readers understand the broader conclusions being drawn from the personal accounts.
- RESPONSE: We attempted to summarize the main points in the text above the quotations. We hope this is clear, but are open to flipping the order if the Reviewer and Editor think this may be better.
- The paragraph starting with "Additionally, at the initial stages of the pandemic..." suggests that service disruptions were quickly addressed. Still, it's unclear how long these disruptions lasted or how they were addressed.
- RESPONSE: We have revised this text to make it clear that measures taken to deal with the pandemic caused some disruptions, but attempts were made to adapt and fix them before they had a significant impact. (Lines 278-279)
- Similarly, statements like "many people in the community did not believe there was a pandemic" could benefit from quantification. Was this the majority of the community or a vocal minority? If this is based on the qualitative interviews, it should be specified.
- RESPONSE: We have clarified that this is a qualitative finding based on interviews with health providers, and thus it cannot be quantified.
- Conclude the Results section with a brief summary statement that encapsulates the key findings, tying the quantitative and qualitative data together.
- RESPONSE: We have included this in the opening paragraph of the discussion. (Lines 297-305)
DISCUSSION
- The discussion about the delivered Figure 4, Penta 1, and Penta 3 vaccines is somewhat confusing. Simplify or break it down further to make it more understandable for readers unfamiliar with the topic.
- RESPONSE: We have revised the wording to make it clearer. (Lines 334-338)
- The limitations section is important and should be highlighted appropriately. Consider starting a new paragraph with a statement, "There are several limitations to this study that need to be addressed."
- RESPONSE: We have made this revision as per the suggestion. (Lines 403-404)
- The conclusion is tucked at the end without a clear break or heading. Consider starting a new paragraph with a statement like "In conclusion," to clearly demarcate it from the rest of the discussion.
- RESPONSE: We have made this revision as per the suggestion. (Lines 415)
- Discuss the practical implications of your findings. How might policymakers, healthcare professionals, or communities use this information?
- RESPONSE: Our findings suggest that there may be a service lag as a result of the pandemic, thus stakeholders at various levels should apply different ways to catch up. Although our study is not in a position to provide firm recommendations, we have attempted to set out some ways that others have taken, which stakeholders at various levels may intend to pursue in future planning. (Lines 385-397)
- While the study's strengths are mentioned, elaboration on why using multiple data sources and having a rapport with the health center are significant. This will bolster the credibility of your study.
- RESPONSE: We have explained why we believe they are the study's strengths. (Lines 399-402)
- There are some minor grammar and phrasing issues, e.g., "...noted for some services the level of effect varied by the regions COVID-19 infection and caseload [20]." might be clearer as "...noted that for some services, the level of effect varied based on the region's COVID-19 infection rates and caseload [20]."
- RESPONSE: We have made this revision as per the suggestion. (Lines 357-359)
- Provide Potential Solutions: The section on regaining momentum after the pandemic would benefit from more detailed, actionable solutions or recommendations.
- RESPONSE: We attempted to highlight several approaches that have succeeded in other contexts, but further implementation study is needed to determine what will work in this context. (Lines 394-397)
BIBLIOGRAPHY
- Update the bibliography. The obsolesce index of the paper is 3 years ( publication date – median references), which is quite high for a topic so actual like this (the median of your references is 2020) , Do an additional bibliographic search and incorporate the latest references on trends and effect of covid in vaccination in the world, Africa, and Ethiopia.
- RESPONSE: We have added extra references to back up the information we included. It's worth noting that many of these references are from 2020-2022 due to the peak of COVID-19 and the subsequent surge in publications during that time.

Round 2
Reviewer 2 Report
The revision of this manuscript has been responsive to the previous review and it has been improved accordingly. Here are a couple of items to attend to.
First, in the text at the top of p. 5, note that if the equation for the specification of the statistical model for the analysis is not separated out from the verbal text in equation format as currently done, then empty space before the words after the equation "Where Y-subt... " should be eliminated. Also, the first letter of "Where" does not need to be capitalized if it is part of the text of the paragraph although a comma after the epsilon-subt would be appropriate. And delete the word "continuous" before variable in the next sentence, as this is conjures up the mathematical term "continuous variable" which connotes a real number in mathematics.
Second, regarding the graphs in Figures 3 and 4, it would be good to include a note as part of the titles of the figures stating that the vertical dashed lines indicate month 19 (March 2020), the month the first case was reported. This will facilitate reading and understanding of the graphs for readers without clarifying this date by checking the text.
The manuscript needs only minor editing as for any journal.
Author Response
Responses to Reviewers
Thank you for taking the time to review our paper. We appreciate your comments and suggestions, which helped us refine our manuscript further. Below we list your comments (in bold) followed by our responses in blue font color.
First, in the text at the top of p. 5, note that if the equation for the specification of the statistical model for the analysis is not separated out from the verbal text in equation format as currently done, then empty space before the words after the equation "Where Y-subt... " should be eliminated. Also, the first letter of "Where" does not need to be capitalized if it is part of the text of the paragraph although a comma after the epsilon-subt would be appropriate. And delete the word "continuous" before variable in the next sentence, as this is conjures up the mathematical term "continuous variable" which connotes a real number in mathematics.
We appreciate this suggestion we have edited as suggested Lines 155-159.
Second, regarding the graphs in Figures 3 and 4, it would be good to include a note as part of the titles of the figures stating that the vertical dashed lines indicate month 19 (March 2020), the month the first case was reported. This will facilitate reading and understanding of the graphs for readers without clarifying this date by checking the text.
We appreciate this suggestion we have added as suggested.